# For Generated Text, Is NLI-Neutral Text the Best Text?

**Michail Mersinias**
Department of Computer Science
The University of Texas at Austin
mmersinias@utexas.edu

**Kyle Mahowald**
Department of Linguistics
The University of Texas at Austin
mahowald@utexas.edu

## Abstract

We explore incorporating natural language inference (NLI) into the text generative pipeline by using a pre-trained NLI model to assess whether a generated sentence entails, contradicts, or is neutral to the prompt and preceding text. First, we show that the NLI task is predictive of generation errors made by GPT-3. We use these results to develop an NLI-informed generation procedure for GPT-J. Then, we evaluate these generations by obtaining human annotations on error types and overall quality. We find that an NLI strategy of maximizing entailment improves text generation when the nucleus sampling randomness parameter value is high, while one which maximizes contradiction is in fact productive when the parameter value is low. Overall, though, we demonstrate that an NLI strategy of maximizing the neutral class provides the highest quality of generated text (significantly better than the vanilla generations), regardless of parameter value.

## 1 Introduction

A perfectly informative Gricean agent (that is, one who strives to be maximally relevant to their interlocutor, as in Grice, 1975) would eschew utterances that are redundant or which contradict that which they have already said. Merrill et al. (2022) give a theoretical argument that text generated by a perfectly informative agent allows for the learning of semantic entailment information.[1] A prediction from this Gricean approach is that most of what speakers say is not entailed by what came before it (since that would not be very informative), nor would it be contradicted by what came before it.

Our goal here is to explore this idea using LLMs, asking whether generated text classified as neutral on a Natural Language Inference (NLI; Conneau et al., 2018; Williams et al., 2018; Bowman et al., 2015) task is less likely to show various kinds of

---

[1]Our code and results are available at https://github.com/Michael-Mersinias/nli_text_generation.

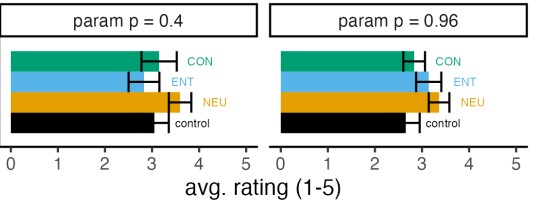

Figure 1: Average holistic ratings for generations from vanilla GPT-J (control), vs. NLI STRATEGIES of maximizing for neutral, contradiction, or entailment, for 2 different choices of parameter values. Neutral performs best in all cases (significantly better than control), but maximizing contradictions is better than the control when randomness is low, and maximizing entailment is better than the control when randomness is high.

errors in text generation, using an LLM like GPT-3. We also ask whether an NLI task could then be used to improve generation. We do not aim for a performant state-of-the-art generation system: the quality of GPT-4 generations are likely higher and more efficient than those here (which involve multiple NLI judgment tasks). Rather, we seek to improve our basic theoretical understanding of the properties of text generated by LLMs, and how they relate to theoretical questions about the emergence of semantic information from text alone.

The NLI task takes a premise and a hypothesis and queries whether the hypothesis is entailed by the premise, contradicts the premise, or is neutral with respect to the premise (Nie et al., 2020; Bowman et al., 2015). We propose that we can use these models, and the extensive body of work around them, in order to guide model generation—essentially using NLI ratings as a way of reranking (Shen et al., 2019; Holtzman et al., 2018) the "fast" output of an LLM, using a well-established logical inference task.

In this vein, Holtzman et al. (2018) used an NLI approach, alongside other approaches, to try to get RNNs to generate more coherent text. Specifically, they focused on selecting sentences that maximize

| $p$ (nucleus sampling param) | CON | ENT | NEU |
|---|---|---|---|
| .40 | 3.44 | 12.93 | 83.62 |
| .96 | 12.41 | 1.37 | 86.20 |

Table 1: For high and low $p$ parameters in SCARECROW, breakdown of NLI classes for generated text. Neutral is by far the most common in both settings, but entailment is more common than contradiction when randomness is low, and *vice versa* when randomness is high.

| | Low p (0.4) | | | High p (0.96) | | |
|---|---|---|---|---|---|---|
| | **CON** | **ENT** | **NEU** | **CON** | **ENT** | **NEU** |
| All | 3.44 | 12.93 | 83.62 | 12.41 | 1.37 | 86.20 |
| CO | 1.66 | 3.33 | 95.00 | 7.85 | 0.52 | 91.62 |
| OP | 12.50 | 0.00 | 87.50 | 23.07 | 0.00 | 76.92 |
| SC | 25.00 | 0.00 | 75.00 | 20.00 | 0.00 | 80.00 |
| IN | 0.00 | 0.00 | 0.00 | 31.81 | 4.54 | 63.63 |
| RD | 2.22 | 28.88 | 68.88 | 16.66 | 6.66 | 76.66 |

Table 2: Distribution of spans in SCARECROW text marked with each error type (CO: correct, OP: off-prompt, SC: self-contradiction; IN: incoherent, RD: redundant) by at least half of annotators, broken down by NLI class.

the probability of being neutral to what preceded them, but did not conduct a thorough evaluation of whether this was the best strategy. While they showed that their models helped generations overall, the NLI part showed mixed results and, on its own, did not help much. Stasaski and Hearst (2022) show that using an NLI task (and particularly using contradictions) can be taken as a measure of semantic diversity. Song et al. (2020) use an NLI task to increase the consistency of the persona in generative models. Nye et al. (2021) use this insight to develop a system that combines LLM generation with a symbolic reasoning model, essentially using the LLM as a fast generation system and then using the symbolic reasoner to refine the output.

We investigate how NLI task judgments relate to text generation errors by systematically investigating whether an NLI task can predict the kinds of errors made by GPT-3 in the publicly-available-for-research SCARECROW dataset (Dou et al., 2022). Since generated text has different failure modes depending on the parameters (i.e. for the nucleus sampling parameter $p$, text generated with high values is more likely to be incoherent/off-prompt, text generated with lower values is more likely to be redundant), we pay particular attention to how the NLI task interacts with the nucleus sampling parameter. We use the results of this analysis to motivate a number of possible ways in which the NLI task can be used to improve generated outputs.

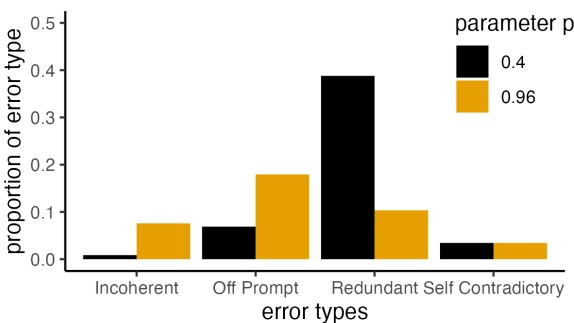

Figure 2: Proportion of erroneous examples in Scarecrow per error type for high and low $p$ parameter.

Comparing across conditions, we generate new text using GPT-J (Wang and Komatsuzaki, 2021) and evaluate it on a subset of the SCARECROW criteria. As proof of concept, we show that using an NLI filter can improve GPT-J generations.

In this work, as in previous work using the NLI task discriminatively, the domain is different than in traditional NLI tasks. In traditional NLI data sets, the premise/hypothesis sentence pairs are typically formulated so as to explicitly stand in a particular logical relation to each other—even when the hypothesis is neutral to the premise. For instance, consider this neutral premise/hypothesis pair from the MNLI homepage: "Your gift is appreciated by each and every student who will benefit from your generosity. / Hundreds of students will benefit from your generosity." While the label is neutral, the sentences stand in a very particular relationship to each other.

In contrast, we assign NLI labels (contradiction, neutral, or entailment) to pairs of sentences that are not explicitly constructed for the task and are thus out-of-domain relative to the NLI training sentences. As such, the NLI labels have different interpretations than they do on the MNLI data sets: e.g., text labeled "entailment" in our data should not be interpreted as "entailment" in the way it would be in an MNLI data set. But, for our purposes, this is not a problem since our goal is to use the NLI models as *tools*. That is, our goal is to take an NLI-tuned model tuned on NLI data (which is specific and idiosyncratic) and apply it to general-purpose text, treating the NLI label as a property of the text that can be used downstream in constraining generations——not because every pair of sentences can be thought of as an NLI task but because, even without that assumption, the NLI labels on sentence pairs are *useful*.

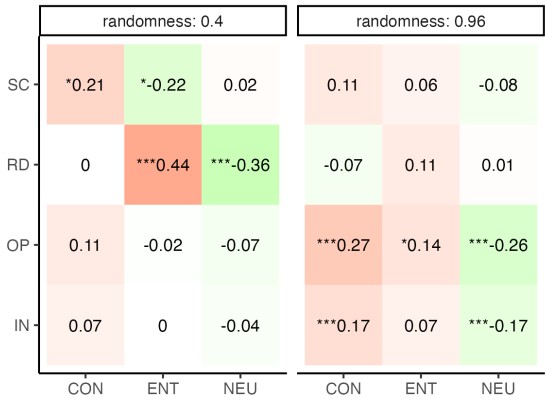

Figure 3: For high and low $p$ (randomness) parameters in SCARECROW, rank correlation between proportion of text showing an error type (y-axis) and the probability of the given NLI class (x-axis).

## 2 Analysis of GPT-3 Text Through Natural Language Inference

First, we conduct a systematic exploration on a subset of the SCARECROW (Dou et al., 2022) dataset (GPT-3 generations with a static temperature parameter value of 1, and either a high or low nucleus sampling parameter $p$ as described). Each dataset entry provides the prompt, the generated text and a list of errors identified by human annotators. For our analysis and evaluation, we focus on "language errors" in SCARECROW , specifically the categories off-prompt (OP), self-contradiction (SC), incoherent (IC), and redundant (RD) error types, as assigned by at least 50% of human annotators.

For attaining NLI ratings, we use an off-the-shelf pre-trained `BART-large-mnli model` for natural language inference. We treat the SCARECROW prompt as the premise and the GPT-3-generated text as the hypothesis and run the NLI task. The distribution of NLI ratings appears in Table 1.

We present the distribution of error types across each NLI class in Table 2. It shows that Correct (CO) segments are very likely to be neutral (95% of the time for low randomness, 92% of the time for high randomness). But, when there is a redundancy (RD) error, the probability of entailment goes way up: to 29% in the low randomness condition and 7% in the high randomness condition (although in all cases, the neutral class remains dominant). When there are off-prompt or self-contradictory errors, in both settings, the probability of the contradiction class increases sharply.

As shown in Figure 3, we computed Spear-

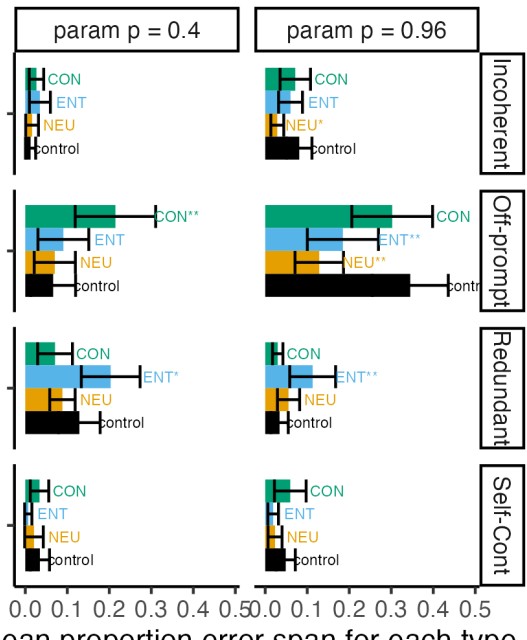

mean proportion error span for each type

Figure 4: For our text generation task, the average human annotator ratings for each of 4 SCARECROW error types, broken up by whether we use vanilla GPT-J output (control), maximize neutral NLI relationships in generated text, maximize entailments, or maximize contradictions. Maximizing neutral is best overall, but maximizing entailment is better than maximizing contradiction when randomness is high and *vice versa* when randomness is low.

man rank correlations between the proportion of text marked with each error type and the NLI probability assigned to each category (entailment/neutral/contradiction). In the low randomness setting, the contradiction NLI class was significantly associated with more self-contradictions and the entailment category with fewer self-contradictions but more redundancy, and the neutral category with less redundancy. In the high randomness setting, the contradiction NLI class was significantly associated with more off-prompt and incoherent errors, entailments with more off-prompt errors, and neutral with *fewer* off-prompt and incoherent errors. All other correlations were not significant at $p < .05$.

Overall, we observe that a low randomness setting leads to a higher probability of text classified as entailment, and that such text is also more likely to be redundant. In contrast, a high randomness setting leads to a higher probability of text classified as contradiction, which is also more likely to be off-prompt or incoherent. In both settings,

text with no errors is significantly more likely to be classified as neutral—lending support to the idea that the neutral class is preferable.

## 3 Realtime NLI to Improve Generation

**Method**    Motivated by this finding, we propose an approach for overcoming the issues present in the generated text of LLMs in order to improve its quality by incorporating *natural language inference* in the text generation pipeline. For text generation, we use the open-source GPT-J (Wang and Komatsuzaki, 2021) while for NLI we use a pre-trained BART-large-mnli model, as above.

Using a random subset of 50 prompts contained in the SCARECROW dataset, we generate a continuation for each of 2 nucleus sampling parameters $p$ (0.4 or 0.96) x 4 conditions (one for vanilla GPT with no NLI constraints, and one for each of our three NLI STRATEGIES as described below), for a total of 8 continuations for each of the 50 prompts (n=400 total).

For the vanilla GPT-J setting, we generate using the relevant parameter for nucleus sampling (0.4 or 0.96), with a max length of 256 tokens including the initial prompt. For the NLI STRATEGIES, we first generate a continuation with a maximum length of 128 tokens including the initial prompt.

Then, the chosen NLI STRATEGY is applied to each candidate sentence of the generated text. The candidate sentence is treated as the "hypothesis", while each sentence of the prompt is used as the "premise". The candidate sentence is appended to the continuation (and therefore becomes part of the hypothesis for subsequent candidates) *iff* it satisfies the chosen NLI STRATEGY for every sentence in the preceding text. Otherwise it is discarded along with the remaining candidate sentences. The NLI STRATEGY conditions are:

- **ENT**: *P(entailment) > P(contradiction)*
- **CON**: *P(contradiction) > P(entailment)*
- **NEU**: *P(neutral) > 0.85*

For instance, if we are applying the **NEU** strategy, we reject a sentence assigned neutral probability < 0.85, relative to *any* sentence in the prompt *or any* previously generated sentence appended to the continuation. We also reject all sentences after it. The process is repeated until there are seven *consecutive* failed attempts to expand the continuation or until the continuation exceeds 256 characters and has 3 or more sentences.

As discussed in the Introduction, our sentences are typically out-of-domain to the NLI task. As such, many sentence pairs of interest to us would be rated neutral. So these NLI STRATEGIESwere chosen so that only pairs with *neutral* probability greater than 0.85 are rated **NEU**. There is no such threshold for the strategies **ENT** or **CON**—which are determined purely by comparing whether the *entailment* probability is greater than the *contradiction* probability or *vice versa*. There are, of course, other ways that we could have operationalized these strategies but we found that these gave us the kind of varied behavior we desired (whereas a strategy that, e.g., only called pairs **CON** if the *contradiction* probability was greater than 0.85 would rarely have been triggered).

In order to ensure sufficient length of generated text for our evaluation, we restarted the process from the initial prompt for any examples whose produced text included less than 2 sentences. After running this process twice, in all cases, at least one sentence was generated that passed the NLI check. See Appendix A for details on the compute set-up.

Two annotators (both undergraduate students at our institution, paid $15 per hour and informed of how the data would be used) evaluated the generations using the SCARECROW annotation framework, as well as a 1-5 holistic rating of overall generation quality (see Appendix B for guidelines). Because some of the prompts were either erroneously not annotated by both annotators or included sensitive content that we removed from our final data set, we ended up with 326 examples with ratings by both annotators and used those for analysis. (The number is relatively small due to a limited budget.) To assess inter-annotator agreement between the two annotators, we measured the correlation of proportion of spans highlighted for each error type across all error types and sentences and found a Pearson correlation of 0.70. Treating error type in a particular category as a binary score, there was 82% agreement between annotators (chance agreement rate would be 56%). Inter-annotator agreement for the holistic ratings were also high, with a Pearson correlation of 0.77 between the two annotators.

**Results**    Focusing first on the average holistic rating assigned to the generation, we observe that maximizing neutral NLI status improves generation. We ran a regression predicting average holistic rating for a prompt, based on the NLI STRATEGY used, with the control (vanilla GPT-J) as the base-

line. When $p$ was 0.4 (low randomness), **NEU** (the neutral strategy) was rated significantly higher than the control ($\beta = .54, p < .05$), as was the case in the high randomness condition ($\beta = .70, p < .01$). As shown in Figure 1, when randomness is low, the **CON** (contradiction strategy) outperforms the control and **ENT** (entailment strategy) underperforms it, but neither significantly so. In the high randomness condition, **ENT** is significantly better than the control ($\beta = .49, p < .01$) but still worse than neutral, while **CON** performs similarly to control.

To better understand the source of the difference, we considered the specific error annotations, as shown in Figure 4. We treated the control as the baseline and ran regressions iteratively for error types and the randomness parameter $p$. For parameter $p = 0.96$ (high randomness), relative to control, **NEU** showed significantly fewer off-prompt errors ($p < .001$) and incoherent errors ($p < .05$). For parameter $p = 0.40$ (low randomness), relative to control, **NEU** did not show significant differences but was particularly better on redundancy errors. When randomness is high, we also observe that **ENT** is significantly better on off-prompt errors but significantly *worse* on redundancy errors. When randomness is low, **CON** is significantly worse for off-prompt errors.

## 4 Conclusion

While the use of an iterative NLI task during generation is likely not practical at scale, we have shown that, as a proof of concept, using NLI as a mechanism for choosing among possible generations seems to be able to imbue text generation systems with a more sophisticated reasoning system (analogous to the idea of using a discriminator to incorporate "System 2" reasoning into generative systems, as described by Nye et al., 2021). In particular, maximizing neutral sentences seems most promising. But it also seems that, in cases where one wants to generate text with a high randomness parameter, maximizing entailments could be productive. Similarly, in a low randomness setting, maximizing contradictions could actually make the text better by avoiding redundancy. As such, this work has implications for understanding the ways in which logical relations among sentences are reflected in text generation using LLMs.

## Limitations

First, while our method incorporates NLI into the text generation process in a real-time and zero-shot fashion, there is still the issue of computational efficiency. Specifically, the NLI STRATEGIES that maximize entailment and contradiction often require multiple generations to produce a sentence which passes their respective NLI checks. Because LLM text generation is already slow for some use cases, this process may cause a bottleneck.

Second, as Nye et al. (2021) show, there is much the NLI task does not capture. NLI tasks capture only a fraction of the possible real-world use cases that one might want to use for guiding generation. Future work might explore using additional kinds of tasks, with the caveat that increasing the complexity of the task could slow generation down even more.

Finally, we tested the generation procedure on only GPT-J, but are open to the possibility that more sophisticated models (especially those like ChatGPT that already include human feedback) might already do better at some of the errors identified in SCARECROW, and so could benefit less from our procedure.

## Acknowledgements

We thank Clara Meister and Jessy Li for helpful conversations. K.M. acknowledges funding from NSF Grant 2104995.

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

## A   Appendix: Compute

The experimental evaluation was conducted in Google Colab Pro+ (Python 3.8.10, Pandas 1.3.5, Numpy 1.21.6, Transformers 4.25.1, Torch 1.13.1+cu116).

We used off-the-shelf models from HuggingFace for our research purposes, consistent with their intended use.

We make our code and data publicly available for research use.

For a single pass of the 326 examples, 35 compute units from an A100-SXM4-40GB GPU and about 3 hours were required.

## B   Appendix: Annotation Guidelines

Below, we reproduce our annotation guidelines and in Figure 5 the SCARECROW guidelines given to our annotators.

**Dataset**   In this research regarding natural language inference (NLI) application in text generation, we have made use of 50 prompts in order to generate 400 pieces of text. For the purpose of text generation, we have used the open-sourced GPTJ model and the nucleus sampling decoding strategy. There are totally eight different combinations of NLI conditions and nucleus sampling parameter values.

**Error Types**   For each example, an annotation has to be completed in accordance with the SCARECROW annotation guidelines. The error types of interest are the "redundant", "off-prompt", "self-contradiction" and "incoherent" ones. An example may have zero, one or multiple error types assigned to it. Furthermore, a rating (1 - 5 stars) should be given to each example in order to represent the overall quality of text generation according to the annotator's judgment. A description of each error type as it is defined by SCARECROW, along with a corresponding example, is presented in Figure 5.

**Annotation Process**   The goal of the annotation process is to explore the quality of the text gener-

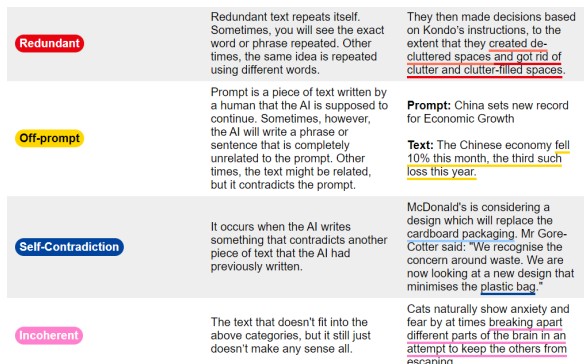

Figure 5: Description of redundant, off-prompt, self-contradiction and incoherent error types along with corresponding examples, according to the SCARECROW annotation framework.

ation which resulted from the respective prompt. Each example consists of the prompt, a -|GEN|- separation token and the GPTJ generated text. Thus, it is of the form: <PROMPT> -|GEN|- <GENERATION>. In order to perform the annotation of an example, the procedure is as follows.

1. Select an error type, then perform a drag selection in the sequence of words that corresponds to it.

2. Repeat the process as needed. An example may have zero, one or multiple error types assigned to it. An error type may also be assigned multiple times.

3. Finally, assign a rating (1 - 5 stars) in order to represent the overall quality of text generation.

Furthermore, if a mistake is made, the "Delete all Regions" button may be selected in order to redo the annotation task for this example. Finally, when the annotation task for an example is complete, the blue "Submit" button should be selected on the top right in order to register the annotation.