# OpenReview forum: "For Generated Text, Is NLI-Neutral Text the Best Text?"
_EMNLP/2023/Conference — EMNLP 2023 Findings_

### Official Review · Reviewer_gehN · 2023-07-19

**Soundness:** 4

**Excitement:**

3: Ambivalent: It has merits (e.g., it reports state-of-the-art results, the idea is nice), but there are key weaknesses (e.g., it describes incremental work), and it can significantly benefit from another round of revision. However, I won't object to accepting it if my co-reviewers champion it.

**Paper Topic And Main Contributions:**

The paper revisits the link between redundancy and statements not being concordant. While this has been studied in NLP already, this paper explores it again with focus on different sampling strategies and the associated decoder scores. Specifically, the authors design a rejection mechanism based on NLI predictions, which they show, through small-scale human evaluation, to reduce errors in the generated text.

**Questions For The Authors:**

Q1: For the NEU scenario, how do you prevent the model from generating completely unrelated sentences (i.e. not even the same topic)?

Q2: For the ENT/CON scenarios, can't it happen that the (ent, con, neu) probabilities are e.g. $(0.11, 0.8, 0.09)$? If this happens, then it is accepted as "entailment" but likely it is a neutral continuation. What was the rationale for simply setting the condition for each ENT/CON/NEU to be $\text{arg}\max p$ = ENT/CON/NEU?

**Reasons To Accept:**

The paper is written very clearly, is placed well within existing literature and has a clear and well-described experiment design. It also shuns away from pursuing the SotA and instead focuses on more fundamental links between text generation and information theory. I believe that as it is, the paper is a good contribution to the community.

**Reasons To Reject:**

While the execution is great, the novelty of this work is disputable as most of the ideas and conclusions could be synthesized from existing works. That said, I do not believe that this should be a reason to reject this paper.

**Reproducibility:**

5: Could easily reproduce the results.

**Reviewer Confidence:**

2: Willing to defend my evaluation, but it is fairly likely that I missed some details, didn't understand some central points, or can't be sure about the novelty of the work.

**Typos Grammar Style And Presentation Improvements:**

The writing is excellent. I only feel like the title is too vague and could be made more concrete whilst retaining its brevity.

---

> ### Author Rebuttal · Authors · 2023-08-25
>
> Thanks for this review, which we found helpful.
>
> *Q: "While the execution is great, the novelty of this work is disputable as most of the ideas and conclusions could be synthesized from existing works. That said, I do not believe that this should be a reason to reject this paper."*
> A: We’re glad the Reviewer appreciates the execution and recommends acceptance. We do not disagree with the claim that many of the ideas here could indeed be synthesized from earlier work. But we do think that the particular framing and execution here is novel in how we are thinking about the relationship between NLI data and “normal text”. Given ongoing interest on the relationship between logical entailiment tasks (see, for instance, Merrill et al. on learning entailment relationships from perfectly communicative text), we think this could be a useful theoretical and empirical contribution.
>
> *Q: "For the NEU scenario, how do you prevent the model from generating completely unrelated sentences (i.e. not even the same topic)?"*
> A: The model is still generating conditional on the existing prompt and sentences so typically doesn’t go too far off the rails, even in the NEU and CON scenarios.
>
> *Q: "For the ENT/CON scenarios, can't it happen that the (ent, con, neu) probabilities are e.g. (0.11, 0.8, 0.09)? If this happens, then it is accepted as "entailment" but likely it is a neutral continuation. What was the rationale for simply setting the condition for each ENT/CON/NEU to be argmax p = ENT/CON/NEU?"*
> A: Yes, good point. The distribution of text generation by LLMs is different than the distribution of the MNLI dataset that the BART model for NLI was trained on, specifically there is an increased probability of Neutral compared to the artificially balanced NLI data. This is shown in Table 1. As we explained in responses to R1, our goal was to use the NLI-tuned classifier to generate useful judgments about naturalistic text (not just NLI text), and simply classifying the vast majority of text as Neutral would not be useful. Rather, we wanted more balanced categories and so designed a thresholding approach that would create a more balanced set. Note that the thresholding procedure was decided independently from the annotations. So, for us, the ENT/CON/NEU labels have a slightly different (but related) meaning than they do in an NLI task.
>
> Responding to these reviews has been useful. We are glad that the reviewers largely appreciated the somewhat different perspective we take on using the NLI task to constrain generations. But we think we can make this theoretical approach even clearer for the Camera Ready.

---

### Official Review · Reviewer_GiAa · 2023-08-04

**Soundness:** 4

**Excitement:**

2: Mediocre: This paper makes marginal contributions (vs non-contemporaneous work), so I would rather not see it in the conference.

**Paper Topic And Main Contributions:**

This paper investigated the impact of different p-value in nucleus sampling. The findings are that high probabiltiy causes contradiction while low p-value causes redundant. Further, the authors designed real NLI to improve generation. The results show the gain of the proposed method.

**Reasons To Accept:**

1 Classify the different errors caused by different probabiltiy settings.
2 By using nature langauge inference module in pipeline, they improve the generation.

**Reasons To Reject:**

Only 50 sampled are evaluated, thus the reliability is not good.

**Reproducibility:**

3: Could reproduce the results with some difficulty. The settings of parameters are underspecified or subjectively determined; the training/evaluation data are not widely available.

**Reviewer Confidence:**

4: Quite sure. I tried to check the important points carefully. It's unlikely, though conceivable, that I missed something that should affect my ratings.

**Typos Grammar Style And Presentation Improvements:**

A performance comparesion between using NLI and not using should be given in table or figure.

---

> ### Author Rebuttal · Authors · 2023-08-25
>
> Thanks for the review!
>
> *Q: "Only 50 sampled are evaluated, thus the reliability is not good."*
>
> A: There were 50 prompts used in order to generate 400 examples, and the evaluation was done on an a subset of those examples, with exclusions as described. To deal with the small data size, we were careful to conduct inferential statistics. Specifically, Figure 1, Figure 3 and Figure 4 display a significant effect, with 95% confidence intervals for Figures 1 and 4, and with significant p-values in Figure 3. Thus, while we agree the experimental setup could ideally benefit from a larger sample size, we were constrained by having to get trained annotators to annotate and think there is enough useful signal here to be informative, based on the statistics. We also note that reliability is certainly correlated with sample size in the population, but it is certainly statistically possible for relatively small samples to be reliable.
>
> *Q: "A performance comparesion between using NLI and not using should be given in table or figure."*
>
> A: The performance of vanilla generations without NLI is annotated as "Control" in Figure 1 and Figure 4, where it it shown that methods using NLI appropriately outperform Control significantly. [Apologies if we misunderstood this query and the Reviewer was asking for something else!]

---

### Official Review · Reviewer_XZPF · 2023-08-05

**Soundness:** 3

**Excitement:**

3: Ambivalent: It has merits (e.g., it reports state-of-the-art results, the idea is nice), but there are key weaknesses (e.g., it describes incremental work), and it can significantly benefit from another round of revision. However, I won't object to accepting it if my co-reviewers champion it.

**Paper Topic And Main Contributions:**

This paper investigates the relationship between NLI relations and the quality of machine-generated text. The authors find that neutral NLI relation between generated text and input prompt is often an indicator of high quality. Contradiction and entailment relations might indicate high quality under low and high-randomness sampling strategies, respectively.

**Questions For The Authors:**

A. What are the significance levels in Figure 3?

**Reasons To Accept:**

Using NLI to guide language generation is an interesting idea. The correlation between NLI relations and the quality of generated text is significant in the experiments. The paper is well-written.

**Reasons To Reject:**

I'm not sure to what extent the findings in this paper are generalizable to all language generation tasks. As the authors pointed out in the Limitation section, it seems impossible to justify that the three categories of NLI relations are sufficient to cover all possible relations between generated text and prompt. In this sense, it is important to include discussions on the generalizability of the method and why the chosen dataset (i.e., SCARECROW) is suitable for this study.

In addition, the reliability of the NLI classifier is also a concern.
Specifically, how many examples are there that cannot be categorized into any of the three categories? How many of them are misclassified as 'neutral'? This matters because a major claim of this paper is that 'neutral' is an indicator of high quality.

**Reproducibility:**

4: Could mostly reproduce the results, but there may be some variation because of sample variance or minor variations in their interpretation of the protocol or method.

**Reviewer Confidence:**

4: Quite sure. I tried to check the important points carefully. It's unlikely, though conceivable, that I missed something that should affect my ratings.

---

> ### Author Rebuttal · Authors · 2023-08-25
>
> *Q: "The three categories of NLI relations are not sufficient to cover all possible relations between generated text and prompt….The reliability of the NLI classifier is also a concern. Specifically, how many examples are there that cannot be categorized into any of the three categories? How many of them are misclassified as 'neutral'?"*
>
> A: This Reviewer raises an important theoretical point that we will aim to clarify in a revision. The Reviewer is exactly right that most sentence pairs are not like the simple pairs typically used in NLI tasks. Our claim does not depend on a close match between the NLI training examples and the test examples. Rather, the idea is to use the NLI models as tools. So, if the text labeled “entailment” was only sometimes entaliment, but it *was* consistently redundant, that would still be informative. In that sense, even the accuracy of the NLI task on these sentences is not crucial: what matters is that the ratings are useful for constraining generation.
>
> In other words, the task is to take an NLI-tuned model tuned on NLI data (which is specific and idiosyncratic) and apply it to general text—not because every pair of sentences is equivalent to an NLI task but because, even without that assumption, we think the NLI-tuned data is useful.
>
> Writing this explicitly would, we think, help clarify the theoretical goals of the paper and we will add a similar discussion for the Camera Ready. Thank you for this review: we think doing this will help clarify our position.
>
> *Q: "Significance levels in Figure 3?"*
>
> A: In Figure 3, we display Spearman Rank Correlations where color represents correlation magnitude and significance is represented by the number of stars which correspond to p values. Specifically, one star corresponds to p < 0.05, two stars correspond to p < 0.01, and three stars correspond to p < 0.001.

---

### Meta-Review · Area_Chair_oPzs · 2023-09-19

**Recommendation:** 3

**Metareview:**

This paper explores leveraging a pretrained NLI model to assess the entailment relationship between generated text, and the prompt and the preceding text. While all reviewers appreciated the execution of the paper, some reviewers expressed concerns regarding the generalizability of the proposed method.

---

### Decision · Program_Chairs · 2023-10-07

**Decision:**

Accept-Findings

**Comment:**

This paper explores leveraging a pretrained NLI model to assess the entailment relationship between generated text, and the prompt and the preceding text. While all reviewers appreciated the execution of the paper, some reviewers expressed concerns regarding the generalizability of the proposed method.